# Livestock grazing is associated with seasonal reduction in pollinator biodiversity and functional dispersion but cheatgrass invasion is not: Variation in bee assemblages in a multi-use shortgrass prairie

Khum Bahadur Thapa-Magar[1], Thomas Seth Davis[1]*, Boris Kondratieff[2]

1 Department of Forest and Rangeland Stewardship, Warner College of Natural Resources, Colorado State University, Fort Collins, Colorado, United States of America, 2 Agricultural Biology, College of Agricultural Sciences, Colorado State University, Fort Collins, Colorado, United States of America

* seth.davis@colostate.edu

**Data Availability Statement:** Data for bee abundances have been uploaded to the Dryad Digital Repository (doi:10.5061/dryad.cjsxksn4w).

## Abstract

Livestock grazing and non-native plant species affect rangeland habitats globally. These factors may have important effects on ecosystem services including pollination, yet, interactions between pollinators, grazing, and invasive plants are poorly understood. To address this, we tested the hypothesis that cattle grazing and site colonization by cheatgrass (*Bromus tectorum*) impact bee foraging and nesting habitats, and the biodiversity of wild bee communities, in a shortgrass prairie system. Bee nesting habitats (litter and wood cover) were marginally improved in non-grazed sites with low cheatgrass cover, though foraging habitat (floral cover and richness, bare soil) did not differ among cattle-grazed sites or non-grazed sites with low or high cheatgrass cover. However, floral cover was a good predictor of bee abundance and functional dispersion. Mean bee abundance, richness, diversity and functional diversity were significantly lower in cattle-grazed habitats than in non-grazed habitats. Differences in bee diversity among habitats were pronounced early in the growing season (May) but by late-season (August) these differences eroded as *Melissodes* spp. and *Bombus* spp. became more abundant at study sites. Fourth-corner analysis revealed that sites with high floral cover tended to support large, social, polylectic bees; sites with high grass cover tended to support oligolectic solitary bees. Both cattle-grazed sites and sites with high cheatgrass cover were associated with lower abundances of above-ground nesting bees but higher abundance of below-ground nesters than non-grazed sites with low cheatgrass cover. We conclude that high cheatgrass cover is not associated with reduced bee biodiversity or abundance, but cattle grazing was negatively associated with bee abundances and altered species composition. Although floral cover is an important predictor of bee assemblages, this was not impacted by cattle grazing and our study suggests that cattle likely impact bee communities through effects other than those mediated by forbs, including soil disturbance or nest destruction. Efforts aimed at pollinator conservation in

**Funding:** Funding was provided by McIntire-Stennis appropriations (USDA NIFA COL 00767) to TSD.

**Competing interests:** The authors have declared that no competing interests exist.

prairie habitats should focus on managing cattle impacts early in the growing season to benefit sensitive bee species.

## Introduction

Wild bees play key functional roles in natural landscapes including the pollination of wild plants and crops and are vital for maintaining biodiversity and ecosystem function [1, 2]. Roughly 90% of the world's plant species are pollinated by animals, in which bees are the dominant flower visitors for pollination services [3]. However, wild bees are declining globally, with serious implications for human food security and ecosystem function [4, 5]. Most authors now agree that wild bees are vital for pollination services in agricultural systems and can exceed the services provided by honey bees (*Apis mellifera* L.) [6, 7]. Accordingly, conservation of wild bee communities is important to maintain pollination services in both agricultural areas and natural landscapes.

Habitat alteration and exotic species introduction are hypothesized to be among the major contemporary drivers directly and indirectly affecting bee communities [8]. In rangeland ecosystems, managed livestock grazing is a dominant process by which habitat alteration occurs [9]. Livestock grazing can impact wild bees directly or indirectly through various mechanisms, including effects on bee nesting and foraging habitats [10] and behaviors [11]. For example, soil compaction due to livestock activity can damage potential or existing ground nesting sites crucial for ground- and cavity-dwelling wild bee species [12] or livestock may consume or alter composition of forbaceous species that wild bees rely on for foraging resources [13, 14]. In addition, livestock may directly kill adult bees as well as their larvae via trampling [15, 16]. Since ground-nesting solitary bee species comprise a substantial proportion of many wild bee communities, these effects are a serious concern for ranch managers concerned with maintenance of ecosystem services and may ultimately affect rangeland productivity. In addition, repeated pressure on plant communities from livestock grazing can also impact plant growth, architecture [17, 18], floral traits, plant attractiveness to pollinators, plant reproductive success [19, 20], and soil characteristics [21]. An understanding of these collective effects on wild bee pollinators in rangelands remains nascent but could be related to functional variation among bee species. For example, it is possible that bee life-history traits (such as below- or above-ground nesting habits) explain the distribution of bee species in grazed- and non-grazed habitats.

In addition to managed livestock grazing, biological invasion is another ecological process driving habitat alteration in rangeland systems and may also have consequences for wild bee communities [1]. Both invasive forbs and grasses affect wild bee communities indirectly through impacts on native plant composition and abundance. Invasive plants may outcompete native forbs for nutrients, light, space and water [22, 23]. Invasive grasses, particularly *Bromus* species including *B. tectorum* L. and *B. japonicus* Thunb. (hereafter, 'cheatgrasses') have colonized many rangeland ecosystems in western North America [24]. Invasion of rangeland habitats in western North America by cheatgrasses is extensive and may impact wild bee communities via multiple mechanisms, but these interactions have not yet been examined. For instance, cheatgrass does not provide food or useful nest-site structures for bees and may gradually replace native forbs by altering disturbance patterns, especially fire cycles [25].

To provide new information on the interactions between pastoral land use, habitat degradation via invasive species, and wild bee communities, we ask the question "How do livestock

grazing (by cattle) and cheatgrass cover impact bee biodiversity?" To answer this question, our objectives were to (1) compare seasonal variation in wild bee assemblages and functional dispersion in rangeland habitats utilized for cattle grazing to non-grazed rangeland habitats with low and high cheatgrass cover; and (2) characterize associations between foraging and nesting resources and bee functional traits. Our studies provide new insights into the relationship between wild bee communities and dominant ecological processes affecting their habitats in a shortgrass prairie ecosystem, with implications for the management of rangelands and maintenance of pollination services.

## Methods and materials

### Description of study area and site selection

Study sites were selected in semiarid shortgrass-steppe habitats in the Front Range region of central- and northern-Colorado (Fig 1). Sites were typically predominated by blue gramma (*Bouteloua gracilis* (Willd. ex Kunth) Lag. ex Griffiths) and buffalo grass (*B. dactyloides* (Nutt.); [26]. The shortgrass-steppe has an evolutionary history of ungulate grazing by bison and elk that predates European settlement. Following European settlement, these rangelands have been managed primarily for cattle grazing [27]. However, thousands of acres of public-domain rangeland areas have been conserved as natural areas, recreational open spaces, or wildlife refuge by state and federal governments. These public lands are typically protected from direct cattle grazing but many have become heavily colonized by invasive species, including cheatgrass [28]. Both public land management agencies and private ranching companies in the region typically use fenced enclosures to control cattle grazing, and we took advantage of existing enclosures to select rangeland study sites that were actively managed for cattle (hereafter referred to as 'grazed' sites, n = 10) and sites where cattle were excluded ('non-grazed' sites, n = 20); in grazed sites mean stocking rates were 93±11 (SE) animal unit months (AUM's). Non cattle-grazed sites (wild ungulates including elk and pronghorn antelope are not excluded from cattle exclosures) were further subdivided to represent locations with low (n = 10) or high (n = 10) cheatgrass cover. All study sites were separated by a minimum distance of 1 km. Permits and permissions for accessing study sites were obtained from multiple agencies

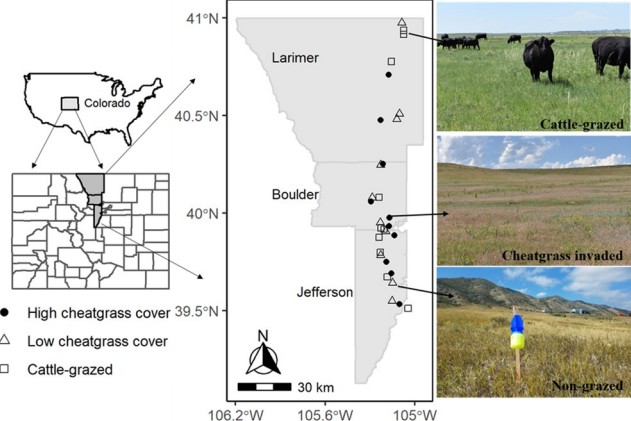

**Fig 1. Map of the study region.** Approximate location of 30 shortgrass prairie study sites distributed across the Colorado Front Range and representative photographs of sites. Study locations were comprised of cattle-grazed sites, sites heavily colonized by cheatgrass (*Bromus* spp.), and sites that were non-grazed and with minimal cheatgrass cover.

including Boulder County Parks and Open Spaces, City of Boulder Open Space and Mountain Parks, City of Fort Collins, and private landowners or ranching companies.

Ground cover was classified using point-intercept transects and used to characterize differences among selected study sites. At each study site, a central point was established from which five equidistant 50 m transects originated; transects were oriented to 0˚, 72˚, 144˚, 216˚ and 288˚ and along each transect an intercept was taken at one meter intervals (250 total intercepts per site). Intercepts at each sample interval were scored as six possible categories including rock, bare ground, wood or litter material, native grass, cheatgrass, or forb. Forbs were further characterized to have either active floral displays (i.e., flowering) or no active floral displays (not flowering or in dry-down). Forbs actively flowering at the time of sampling were also identified in the field to the lowest possible taxonomic level to estimate richness of floral cover. To further account for seasonal variation in bee foraging habitat (floral cover and richness), floral cover sampling was repeated four times during the growing season of 2018 in May, June, July, and August with each sampling occurring mid-month.

From this sampling effort we verified that non-grazed sites were reliably grouped into two categories representing areas of variable cheatgrass cover, and that both cattle-grazed and non-grazed sites had similar forb cover and floral richness (Table 1). Ground cover data and records of seasonal variation in floral cover and richness were subsequently used to evaluate relationships between bee assemblages and habitat factors (described below in *Data analysis* section).

## Bee collection procedures

Bees were collected from each study site using a passive trapping method ('blue vane' traps). Traps consisted of an ultra-violet reflective blue vane fixed to a yellow collection bucket (SpringStar, Woodinville, WA, USA). Although previous research suggests that bee sampling method may impact detection of habitat factors influencing bee communities [29], blue vane traps are well suited for collecting across large landscapes as they are easily deployed and are not biased to observer skill or abilities [30–32]. Traps were placed at the previously established central location at each site to sample bee assemblages over four separate periods (May, June, July, and August) that corresponded with the assessments of floral cover. In each trapping period, traps were hung from wooden stakes at a height of 1 m, and trap contents were collected after 48 h. Bees were collected into plastic bags, placed on dry ice, and immediately returned to the laboratory for curation.

All collected bee specimens were pinned, mounted, sorted to morphospecies and were subsequently identified to the lowest taxonomic level possible, in most cases this was to genus and

**Table 1. A comparison of ground cover and floral richness across grazed and non-grazed rangeland sites.**

| Variable | Site category | | | $F_{(2,27)}$ | $P$ |
|---|---|---|---|---|---|
| | Cattle-grazed | Non-grazed, high cheatgrass cover | Non-grazed, low cheatgrass cover | | |
| **Native grass cover (%)** | **41.4[a]± 2.8** | **13.9[b]±1.2** | **30.7[a]±2.0** | **23.5** | **<0.001** |
| **Cheatgrass cover (%)** | **5.5[a]±1.4** | **26.3[b]±3.5** | **2.1[a]±0.6** | **18.6** | **0.003** |
| Floral cover (%) | 8.2[a]±3.7 | 10.2[a]±3.1 | 10.39[a]±4.1 | 0.2 | 0.890 |
| Litter/wood cover (%) | 12.5[a]±2.2 | 10.7[a]±1.4 | 18.9[a]±2.2 | 2.6 | 0.090 |
| Bare ground cover (%) | 12.3[a]±1.9 | 5.8[a]±1.1 | 8.84[a]±1.4 | 2.4 | 0.110 |
| Rock (%) | 1.1[a]±0.5 | 1.9[a]±0.6 | 0.4[a]±0.2 | 1.3 | 0.291 |
| Floral richness | 16.0[a]±0.7 | 19.0[a]±0.5 | 19.0[a]±0.5 | 0.7 | 0.475 |

Values are mean percentages of cover in each category plus or minus one standard error of the mean, and superscript lettering denotes Tukey's HSD test.

species. Specimen identifications were confirmed by insect taxonomists external to the study [33]. Vouchers of identified bee specimens are curated at the C.P. Gillette Museum of Arthropod Diversity at Colorado State University (Fort Collins, Colorado).

## Bee functional traits

Bee qualitative and quantitative functional traits were compiled for the purposes of calculating functional dispersion, a metric that describes the relative diversity of functional traits in a species assemblage [34]. We considered multiple ecological traits related to wild bee life history, behavior, and foraging ranges including diet breadth (lecty), nesting habit and nest locations, pollen carrying structures, sociality, and body size [35].

Traits including intertegular distance (ITD, a proxy for body size) and tibial hair density were resolved using high-resolution photographic methods as follows: photographs were taken for ten replicate specimens (5 male, 5 female) per species from three orientations (head, dorsal and ventral views) for each of 49 species using Canon-EOS Rebel T7i DSLR (49 species ×3 orientations ×10 specimens per species = 1470 photograph images). For rare species (i.e., those that were represented by less than 5 males and/or 5 females) supplementary specimens were acquired from museum collections (C.P. Gillette Museum of Arthropod Diversity) for photography and trait characterization. ITD was measured from photograph layers using the image J program [36] to generate an average value for each species. For categorical life history traits, we used scientific literature, online databases, books and field observations for traits classification [S1 and S2 Tables; 33, 35, 37–40]. Individuals that were not positively identified to species, but able to be identified to genus, were assigned trait values from the closest congener considered to have a similar life history [35]. Flight phenology (early, middle, or late-season) was assigned based on the collection period in which abundances were maximized for a given species (S3 Table).

## Data analysis

All analyses were implemented in R version 3.6.2 and, unless otherwise stated, incorporate a Type I error rate of α = 0.05 for assigning statistical significance. However, modeled effects were interpreted as marginally significant at the α = 0.10 level. In parametric analyses using continuous variables, response and predictor variables were standardized to meet assumptions of normality and homogeneity.

**Computation of bee diversity indices and functional dispersion.** A bee species abundance matrix was used to derive species-level abundances as well as bee species richness and α-diversity (Shannon's H') for each site × collection date combination. We computed functional dispersion (FDis) for bee assemblages at each site × collection date combination using the methods of Laliberté and Legendre [34] and metrics shown in S2 Table; FDis was computed using the R add-on package 'FD' [41] and applying the Cailliez correction for non-Euclidean distances generated by inclusion of categorical traits. The metrics of bee species abundance, species richness, diversity, and FDis were used as response variables in the analyses described below.

**Objective 1: Analyze the relationships between habitat types and seasonal variation in bee assemblages and functional dispersion.** We examined how cattle grazing or cheatgrass colonization affect bee diversity using several statistical approaches. First, we tested the fixed effects of site classification (n = 3) and collection period (n = 4; May, June, July, and August) and the site classification × collection period interaction on the responses of mean bee abundance, richness, diversity, and FDis using a two-way ANOVA model.

Sampling curves were generated to estimate and compare rates of species detection across the three different site classifications and was implemented using the R add-on package 'iNEXT' [42]. To quantify β-diversity and turnover in genera across collection periods and sample locations, we used nonmetric multidimensional scaling (NMDS) of Bray-Curtis dissimilarities to evaluate variability in bee assemblages across habitats and sample month.

**Objective 2: Characterize associations between foraging and nesting resources and bee functional traits.** We also examined how variation in foraging and nesting resources affected bee community metrics to determine whether efforts to manage cover would have potential impacts on bee assemblages. We used a generalized linear model with an identity link function to analyze variation in bee assemblage abundance, richness, diversity, and FDis due to variation in cover composition (rock, bare soil, wood/litter, grass, cheatgrass, and floral cover) and floral richness.

To analyze the associations between specific bee functional traits and local habitat factors we used fourth-corner analysis [43, 44] implemented in the R add-on package 'mvabund' [45]. Generalized linear models of fourth-corner statistics were fit for bee species abundances as a function of a matrix of species traits and environmental variables (and their 2-way interaction) using a Least Absolute Shrinkage and Selection Operator's (LASSO) penalty which restricts influences of interactions that do not add to the Bayesian Information Criteria (BIC). Analysis of model deviance was estimated using a Monte-Carlo resampling procedure (9,999 resamples) to evaluate the global significance of trait-environment relationships.

## Results

### Objective 1: Analyze the relationships between habitat types and seasonal variation in bee assemblages and functional dispersion

A total of 4,368 bees representing four families (Apidae, Colletidae, Halictidae, and Megachilidae) were captured in blue vane traps. The four families were represented by 18 genera and 49 species. The European honeybee, *Apis mellifera*, represented only ~2% of the total collection, indicating that cultured bees had relatively little impact on the study. Three genera including bumble bees (*Bombus* spp.), long-horned bees (*Melissodes* spp.), and furrow bees (*Halictus* spp.) collectively comprised about 63% of the sample (Table 2). Rarefaction analysis indicated that rates of species detections were similar among the three habitat classifications (S1 Fig).

There were significant differences in bee community metrics due to site classification, month of collection, and their interaction. Bee abundance varied significantly due to the main effects of site classification ($F_{2, 109} = 3.437$, $P = 0.035$) and collection period ($F_{3,109} = 15.785$, $P<0.001$), but there was no evidence of an interaction between these terms ($F_{6, 109} = 0.655$, $P = 0.685$; Fig 2A). On average, bee abundances were 18 and 29% in sites with high cheatgrass cover than in sites with low cheatgrass cover or sites that were cattle-grazed, respectively. Post-hoc tests revealed that this difference was statistically significant and mean bee abundances differed between sites with high cheatgrass cover and cattle-grazed sites, but bee abundances in sites with low cheatgrass cover were intermediate and not statistically different from either category. Average bee captures in June and July were similar and were 66 and 19% higher than captures in May and August, respectively (Table 3a).

Bee species richness also varied due to the main effects of site classification ($F_{2, 109} = 8.431$, $P<0.001$) and collection period ($F_{3,109} = 21.072$, $P<0.001$), but not their interaction ($F_{6,109} = 0.858$, $P = 0.528$; Fig 2B). Post-hoc analysis revealed that captured bee species richness was on average 22% higher in sites with high cheatgrass cover than sites with low cheatgrass cover and cattle-grazed sites (which did not differ from one another). Similar to patterns found for bee

**Table 2. Summary of all bee taxa captured during the study (γ-diversity) and their abundances.**

| Family | Genus | species | Habitat category | | |
|---|---|---|---|---|---|
| | | | cattle-grazed | high cheatgrass cover | low cheatgrass cover |
| Apidae | *Anthophora* | *affabilis* | 37 | 40 | 31 |
| | | *bomboides* | 0 | 5 | 2 |
| | | *montana* | 43 | 50 | 29 |
| | | *occidentalis* | 56 | 26 | 26 |
| | *Apis* | *mellifera* | 18 | 33 | 40 |
| | *Bombus* | *appositus* | 36 | 60 | 59 |
| | | *bifarius* | 1 | 0 | 1 |
| | | *californicus* | 0 | 0 | 2 |
| | | *centralis* | 0 | 2 | 2 |
| | | *fervidus* | 36 | 113 | 104 |
| | | *griseocollis* | 9 | 18 | 19 |
| | | *huntii* | 4 | 28 | 19 |
| | | *insularis* | 0 | 1 | 0 |
| | | *nevadensis* | 46 | 157 | 85 |
| | | *pensylvanicus* | 114 | 197 | 170 |
| | | *rufocinctus* | 2 | 7 | 4 |
| | | *sylvicola* | 2 | 6 | 3 |
| | *Diadasia* | *enavata* | 23 | 2 | 4 |
| | *Eucera* | *hamata* | 14 | 60 | 30 |
| | | *lepida* | 0 | 4 | 0 |
| | *Melecta* | *pacifica* | 3 | 15 | 10 |
| | *Melissodes* | *agilis* | 120 | 49 | 90 |
| | | *communis* | 148 | 131 | 117 |
| | | *coreopsis* | 28 | 16 | 37 |
| | | sp.1 | 6 | 3 | 6 |
| | | *tristis* | 143 | 64 | 65 |
| | *Svastra* | *obliqua* | 45 | 104 | 94 |
| | | *petulca* | 3 | 5 | 10 |
| | *Xeromelecta* | *interrupta* | 5 | 16 | 5 |
| Colletidae | *Colletes* | sp.1 | 2 | 0 | 0 |
| Halictidae | *Agapostemon* | *angelicus* | 24 | 18 | 11 |
| | | *coloradinus* | 7 | 1 | 3 |
| | | *texanus* | 14 | 29 | 25 |
| | | *virescens* | 27 | 37 | 14 |
| | *Augochlorella* | *aurata* | 8 | 14 | 3 |
| | *Halictus* | *halictus.spp* | 20 | 24 | 33 |
| | | *ligatus* | 8 | 9 | 3 |
| | | *tripartitus* | 139 | 114 | 96 |
| | *Lasioglossum* | *dialictus* | 51 | 117 | 64 |
| | *Lasioglossum* | sp.1 | 8 | 12 | 13 |
| Megachilidae | *Anthidium* | sp.1 | 0 | 7 | 6 |
| | *Lithurgopsis* | *apicalis* | 6 | 10 | 10 |
| | *Megachile* | *dentitarsus* | 1 | 1 | 2 |
| | | sp.1 | 2 | 13 | 4 |
| | | sp.2 | 23 | 38 | 25 |
| | | sp.3 | 2 | 0 | 2 |

(*Continued*)

**Table 2.** (Continued)

| Family | Genus | species | Habitat category | | |
|---|---|---|---|---|---|
| | | | cattle-grazed | high cheatgrass cover | low cheatgrass cover |
| | *Osmia* | sp.1 | 6 | 18 | 2 |
| | | sp.2 | 4 | 15 | 1 |
| | | sp.3 | 1 | 1 | 2 |

Values are total abundances of captured bee species across sampled habitat types, pooled across sites.

abundance, species richness in June and July were similar and were on average 53 and 14% higher than in May and August, respectively (Table 3b).

Bee diversity (as measured by Shannon's H' statistic) also varied significantly due to the main effects of site classification ($F_{2, 103}$ = 10.805, $P<0.001$), collection period ($F_{3,103}$ = 21.485, $P<0.001$), as well as their interaction ($F_{6,103}$ = 2.529, $P = 0.025$). Early in the growing season sites with high cheatgrass cover had significantly higher diversity than either cattle-grazed or non-grazed sites (which did not significantly differ from one another), but by later in the growing season, cheatgrass-colonized and non-grazed sites were similar in terms of diversity but diversity significantly declined in cattle-grazed sites (Fig 2C; Table 3c).

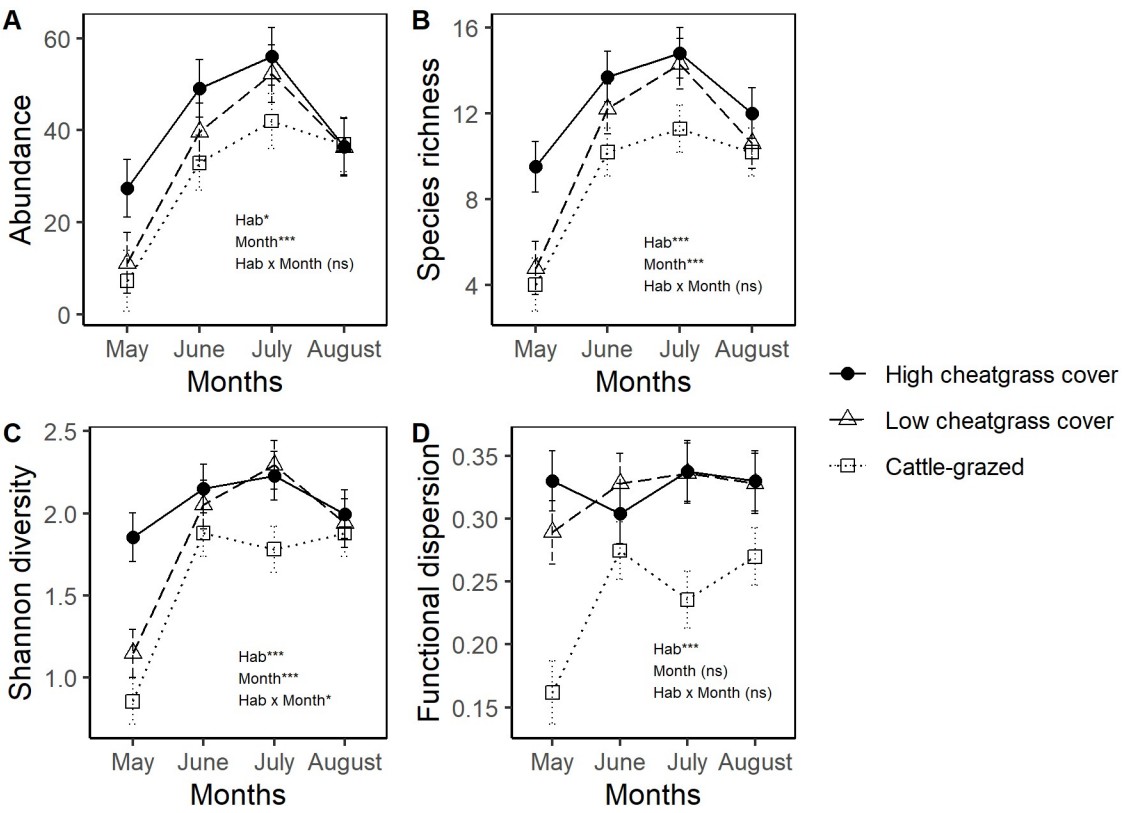

**Fig 2. Bee community metrics vary across grazing treatments and seasonality.** Variation in mean (A) bee abundance, (B) species richness (C) diversity, and (D) FDis represented as a habitat classification × collection period interaction. Asterisks denote significance of main effects (habitat, month of collection) and their interaction, and error bars show ±1 SE of the mean. $^*P<0.05$, $^{**}P<0.01$, $^{***}P<0.001$.

**Table 3. Summary of post-hoc tests comparing bee community metrics across collection period and habitat type.**

| Response variable | Factor | Factor levels | Mean ± SE | Grouping (Tukey's HSD) |
|---|---|---|---|---|
| (a) Bee abundance | Habitat | Cattle-grazed | 29.76 ± 3.06 | B |
| | | Low cheatgrass cover | 34.85 ± 3.17 | AB |
| | | High cheatgrass cover | 42.25 ± 3.12 | A |
| | Month | May | 15.67 ± 3.03 | C |
| | | Jun | 40.32 ± 3.65 | AB |
| | | Jul | 49.83 ± 5.00 | A |
| | | Aug | 36.58 ± 2.25 | B |
| | Habitat × month | n.s. | - | - |
| (b) Bee richness | Habitat | Cattle-grazed | 9.14 ± 0.73 | B |
| | | Low cheatgrass cover | 10.61 ± 0.71 | B |
| | | High cheatgrass cover | 12.50 ± 0.69 | A |
| | Month | May | 6.21 ± 0.77 | C |
| | | Jun | 11.96 ± 0.74 | AB |
| | | Jul | 13.38 ± 0.84 | A |
| | | Aug | 10.90 ± 0.46 | B |
| | Habitat × month | n.s. | - | - |
| (c) Shannon diversity (H') | Habitat | Cattle-grazed | 1.58 ± 0.10 | B |
| | | Low cheatgrass cover | 1.85 ± 0.09 | A |
| | | High cheatgrass cover | 2.05 ± 0.06 | A |
| | Month | May | 1.27 ± 0.13 | B |
| | | Jun | 2.02 ± 0.07 | A |
| | | Jul | 2.09 ± 0.08 | A |
| | | Aug | 1.93 ± 0.05 | A |
| | Habitat × month | Cattle-grazed, May | 0.85 ± 0.24 | C |
| | | Cattle-grazed, Jun | 1.87 ± 0.15 | A |
| | | Cattle-grazed, Jul | 1.78 ± 0.15 | AB |
| | | Cattle-grazed, Aug | 1.87 ± 0.09 | A |
| | | Low cheatgrass cover, May | 1.14 ± 0.18 | BC |
| | | Low cheatgrass cover, Jun | 2.05 ± 0.12 | A |
| | | Low cheatgrass cover, Jul | 2.29 ± 0.08 | A |
| | | Low cheatgrass cover, Aug | 1.93 ± 0.08 | A |
| | | High cheatgrass cover, May | 1.85 ± 0.15 | A |
| | | High cheatgrass cover, Jun | 2.15 ± 0.11 | A |
| | | High cheatgrass cover, Jul | 2.22 ± 0.12 | A |
| | | High cheatgrass cover, Aug | 1.99 ± 0.12 | A |
| (d) Functional dispersion | Habitat | Cattle-grazed | 0.23 ± 0.01 | B |
| | | Low cheatgrass cover | 0.32 ± 0.00 | A |
| | | High cheatgrass cover | 0.32 ± 0.00 | A |
| | Month | May | 0.26 ± 0.02 | A |
| | | Jun | 0.30 ± 0.01 | A |
| | | Jul | 0.30 ± 0.01 | A |
| | | Aug | 0.30 ± 0.01 | A |
| | Habitat × month | n.s. | - | - |

Post-hoc pairwise comparison of means tests (Tukey's HSD test) are shown for all significant factor levels of a two-way ANOVA model analyzing variation in bee abundance, richness, diversity, and functional dispersion. Lettering in the grouping column denotes Tukey's HSD test, and factor levels not connected by the same letter are significantly different at a Type I error rate of α = 0.05. The abbreviation 'n.s.' indicates no significant variation in a response variable due to a modeled effect; therefore, no post-hoc test is shown.

Functional dispersion (Fdis) of bee assemblages varied due to the main effect of site classification ($F_{2,109}$ = 18.266, $P$<0.001) and varied marginally across collection periods ($F_{3,109}$ = 2.539, $P$ = 0.060), but did not vary due to an interaction between collection period and site classification ($F_{6,109}$ = 2.048, $P$ = 0.158, Fig 2D). Bee FDis was significantly reduced in cattle-grazed sites and was on average 28% lower than in non-grazed sites; there was no difference in mean FDis between sites with low- and high cheatgrass cover (Table 3d). Post-hoc tests did not reveal clear pairwise differences in FDis across seasons, though Fdis was on average 14% lower in May than in other summer months (Jun-Aug).

Analysis of bee community composition with NMDS indicated distinct differences in species assemblages between cattle-grazed and sites with high cheatgrass cover, but species assemblages in sites with low cheatgrass cover were similar to both cattle-grazed and high-cheatgrass cover sites (Fig 3A). Differences in species assemblages between cattle-grazed sites and sites with high cheatgrass cover were generally reflected by a turnover in the ratio of *Bombus*: *Melissodes* species; however, abundances of multiple genera were consistent across site classification (Table 2). There were also distinct seasonal differences in the genera composition of bee assemblages with both *Bombus* and *Melissodes* becoming more abundant throughout the season and all other species generally becoming less prevalent (Fig 3B), though some genera such as *Agapostemon* were consistent in their abundances throughout the growing season (S3 Table).

**Objective 2: Characterize associations between foraging and nesting resources and bee functional traits.**  Linear model analysis testing ability of habitat components (cover) to predict variation in bee assemblages revealed that, although elements of foraging or nesting habitat were not strongly differentiated by site classifications, some were nonetheless good predictors off bee community metrics (S4 Table). Specifically, there was significant positive association between bee abundances and floral cover ($\beta$ = 0.549, $P$ = 0.037), although the species richness of bee assemblages was not associated with any cover factor or floral richness. Similarly, diversity of bee assemblages was not significantly associated with any cover factors. However, the FDis of bee communities was significantly negatively associated with increasing bare ground cover ($\beta$ = -0.673, $P$ = 0.007), and FDis was also marginally negatively associated with increasing grass cover ($\beta$ = -0.848, $P$ = 0.066; Fig 4).

Fourth-corner analysis revealed significant patterns in the correlations between habitat characteristics, bee life history traits, and bee species abundances (model deviance = 3.377, $P$<0.001). Bee body size (ITD) was positively associated with floral richness, indicating that captured bees tended to be larger as floral richness increased. Bee nest locations were correlated with habitat classification, and below-ground nesters were more abundant in cattle-grazed and cheatgrass-colonized, whereas above-ground nesters were less abundant in these areas. Diet breadth was also correlated with environmental conditions and oligolectic bees were less abundant when floral cover was high but more abundant with high grass cover, whereas the opposite was true for polylectic species; kleptoparasitic bee abundances were unrelated to cover or habitat classification. Solitary bees were less abundant in areas where floral cover and richness was high but increased in abundance in areas with high grass cover and bare soil, whereas social species were more abundant with increasing floral richness but were negatively associated with grass and bare soil cover. Variation in abundances of kleptoparasitic species and species with flexible social behaviors were not related to cover or habitat classification. Only bee species exhibiting early-season phenologies were impacted by cover, and early-season species were more abundant in areas colonized by cheatgrass. Abundances of bee species also varied due to interactions between pollen collection-related traits and environmental conditions. Bees with scopa pollen collection structures were positively associated with high grass and soil cover but negatively associated with high floral richness and rock cover, whereas

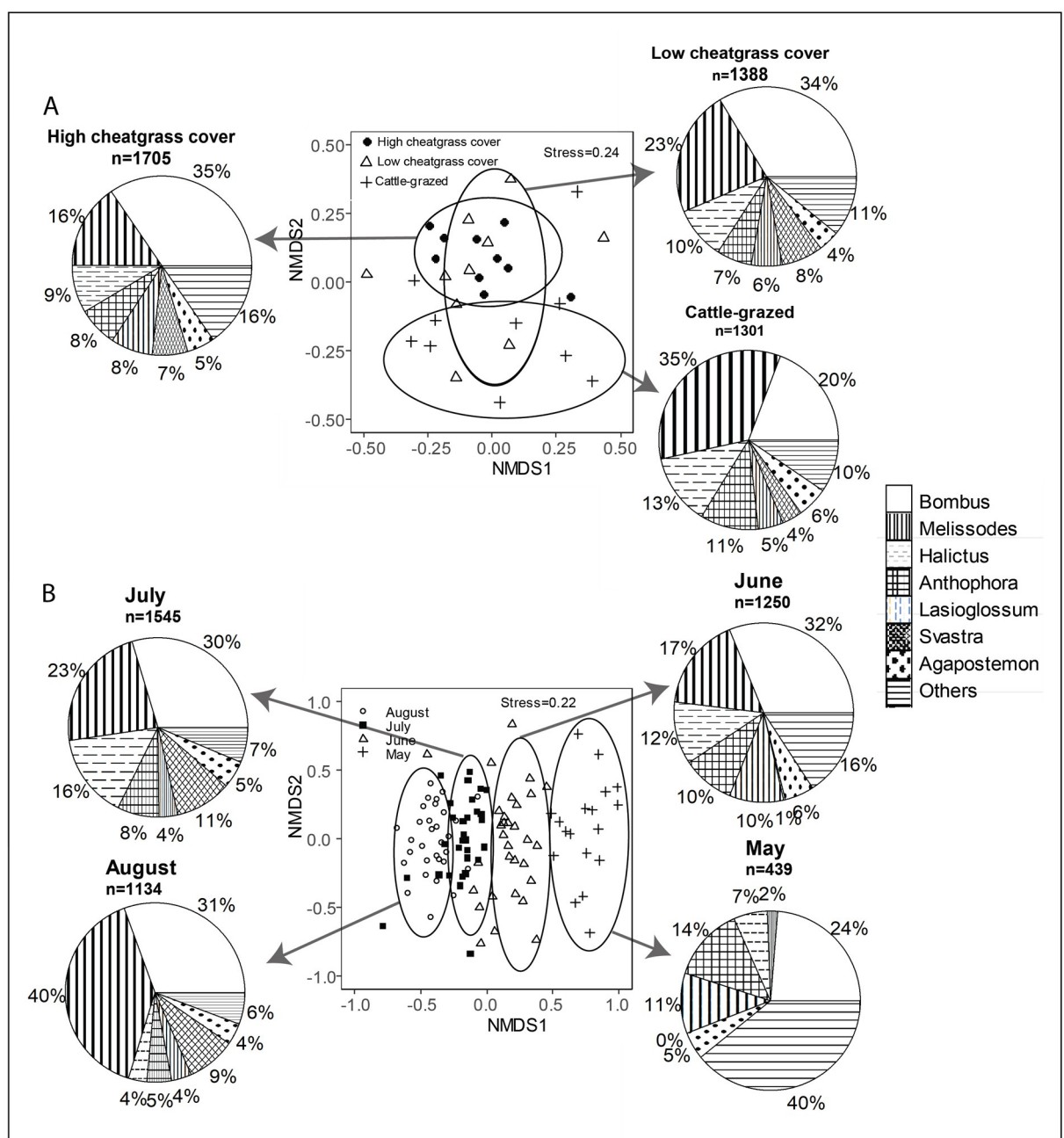

**Fig 3. Variation in bee assemblages in cattle-grazed, cheatgrass-colonized, and non-grazed sites.** Non-metric multidimensional scaling (NMDS) plots of bee assemblages (grouped by genera) relative to (A) habitat classification and (B) collection period.

bees with corbicula were positively associated with high floral richness and rock cover but negatively associated with cheatgrass and bare soil cover. Variation in tibial hair densities had complex relationships with environmental conditions; bees with high tibial hair densities were more abundant in areas with high grass and soil cover, whereas bees with low tibial hair densities were more abundant in areas with high floral richness and rock cover, and bees with intermediate tibial hair densities were most abundant in areas with high floral and cheatgrass cover (Fig 5).

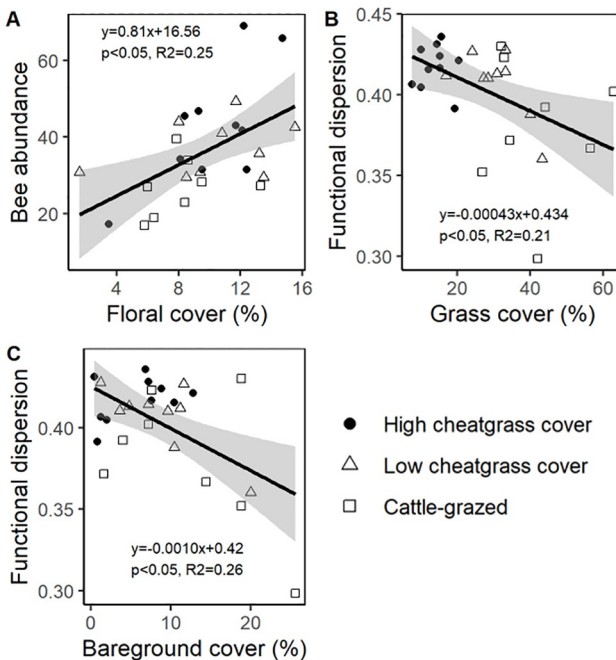

**Fig 4. Ground cover impacts bee abundance and functional dispersion.** (A) Floral cover is associated with increases in bee abundances, but both (B) grass cover and (C) bare soil cover are associated with reduced functional dispersion in bee assemblages. Gray shading shows 95% confidence intervals and regression equations are provided in each panel.

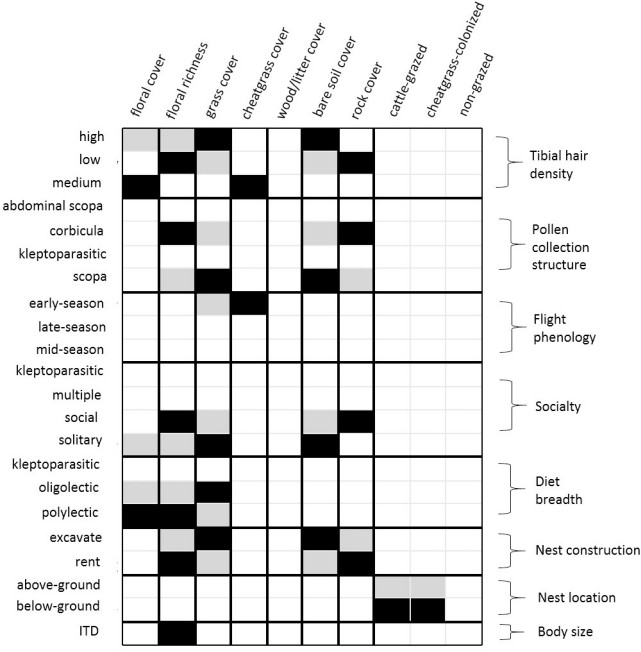

**Fig 5. Relationship between habitat factors and frequency of bee functional traits.** Summary of fourth-corner analysis to model bee species abundances as a function of life history trait × environment interactions. Black cells indicate positive regression coefficients, gray cells indicate negative coefficients. Blank cells indicate no relationship. Identified correlations are significant at $P < 0.10$.

## Discussion

Cattle-grazing and cheatgrass colonization of shortgrass prairie sites were not associated with large differences in bee foraging habitats (floral cover and species richness) but did reflect a difference in wild bee nesting habitats in terms of the proportion of native grass cover and woody material on the ground surface (Table 1). Despite the modest differences in cover composition across habitat classifications our data provide evidence that cattle grazing is associated with significant reductions in early- and mid-season bee diversity and FDis, but this was not the case in non-grazed sites with dense cheatgrass cover (Fig 2). There were distinct differences in community composition between cattle-grazed and non-grazed sites with high cheatgrass cover that was reflected by turnover in the ratio of *Bombus* spp: *Melissodes* spp.; however, bee assemblages in non-grazed sites with low cheatgrass cover were similar to both grazed sites and those with high cheatgrass cover, and were the most variable overall (Fig 3). Collectively, these results indicate that FDis in bee communities is more strongly predicted by broad-scale habitat classification (i.e., cattle-grazed vs. high- or low-cheatgrass cover) than cover composition within specific sites, with potential consequences for pollination services in rangelands.

Landscapes in the study region share a long evolutionary history with bison, elk, and other wild grazing and browsing species [46] and forbs may therefore be well-adapted to tolerate grazing, which could partially explain why no differences in floral cover were observed across site classifications. Nonetheless, floral cover predicted bee abundances with more bees captured from sites with abundant flowering forbs (Fig 4). In other recent studies, locations with high floral density have been associated with fewer bee captures in passive traps (e.g., [29]) due to reduced attractiveness of traps when abundant floral resources are available.

Analysis of bee functional traits relative to floral cover and richness revealed that the preponderance of bees at sites with high floral cover were those with life history traits that included sociality, polylecty, and large body size. In our collections, this combination of traits is mostly represented by bumblebees (*Bombus* spp.). Accordingly, management efforts aimed at increasing or restoring local floral densities may be more likely to benefit *Bombus* spp. than other taxa. Interestingly, both cattle-grazed sites and those with high cheatgrass cover had similar relationships with bee functional traits and were positively associated with higher abundances of bees with below-ground nesting habits (Fig 5). In some landscapes cattle may trample sensitive arthropod species resulting in reduced abundances [47], but this does not appear to be the case for below-ground nesting bees in our system. Although bee abundances did not differ between grazed and non-grazed sites, cattle grazing was associated with significant reductions in bee FDis indicating that cattle presence may result in a loss of bee functional diversity. The mechanisms underlying this pattern merit further study, as pollination services are generally improved with increasing bee functional diversity [48]. Since floral abundance and richness were not negatively impacted at grazed sites, we hypothesize that impacts of cattle on bee assemblage functional diversity are mediated via nesting habitats, rather than through indirect consumption-mediated effects on foraging habitat. In other systems cattle-grazing has been documented to have positive effects on bee abundances even at very high grazing intensities [49], so it may be difficult to generalize cattle-driven effects on bee assemblages.

To our knowledge, this is among the first studies to evaluate the effects of a non-native grass on pollinator assemblages. Our findings suggest that sites with high cover of cheatgrass were not associated with significant reductions in bee abundance, diversity, of FDis; instead, cheatgrass-dominated sites tended to have higher bee abundance and diversity early in the growing season (Fig 2). This contrasts with findings from other recent studies; for instance, Bhandari et al. [50] determined that pollinator abundances in semi-arid pastures were reduced under high densities of non-native forage species. Several non-mutually exclusive hypotheses

could potentially explain this pattern. First, it is possible that at cheatgrass-colonized sites vane traps were more visually apparent due to the relatively homogenous structure of the vegetation and thus more attractive to foraging bees. For example, some authors have suggested that passive traps tend to become increasingly attractive when floral displays are not abundant [29, 32, 51]. Similarly, bees captured in sites with high cheatgrass cover may be responding from nearby patches of foraging habitat or recruited from other distal locations. Alternatively, sites that are occupied by cheatgrass may simply be on highly productive or suitable soils; in other words, highly productive sites may be generally superior for invasive grasses, forbs, and pollinators alike. However, this seems unlikely as floral cover did not differ between site classifications (Table 1), and there was no evidence of a correlation between cheatgrass cover and floral cover (Pearson's $r$ = 0.08). Another possibility is that sites with high cheatgrass cover provide some as-of-yet undetermined benefit to nesting habitat, such that bees are more likely to occupy sites with high cheatgrass cover even if there is little relationship between cheatgrass cover and foraging habitat (floral cover). In future studies it will be important to determine whether the effects of cheatgrass colonization are consistently associated with high early-season bee abundance and diversity and, if so, whether these effects are an artifact of sampling strategy or due to some ecological effect such as improved nesting habitat. Accordingly, our findings do not currently suggest a need to mitigate cheatgrass occurrence for pollinator conservation efforts.

Seasonal variation in wild bee assemblage richness and functional diversity were considerable, and our sample underscores the importance of making collections across the growing season to generate reliable estimates of bee richness and diversity. There was evident turnover in taxa with certain species of *Eucera*, *Melecta*, and *Osmia* prevalent early in the growing season, but by June and July *Bombus*, *Halictus*, *Lasioglossum*, and *Melissodes* were predominant in study sites (Table 2). Altogether, bee taxa richness and diversity were lowest in the early growing season, which is consistent with other reports [52] and was mostly due to the relative inactivity of many social and semi-social species in the spring. Our collection had a lower rate of species detection than other regional studies focusing on bees in Colorado grasslands. For example, Kearns and Oliveras [53] detected 108 species in grasslands of Boulder County, Colorado and an earlier study by Cockerell [54] detected 116 species. This could be due in part to differing collection methods used among the different studies; for instance, blue vane traps, hand netting, and bowl traps are known to differ slightly in terms of the community they sample [29]. In general, bumblebees tend to be slightly biased towards blue vane traps, thought blue vane traps also tend to capture the greatest overall number of taxa. In addition, these earlier studies found that floral resources were generally positively associated with intermediate levels of cattle grazing. In both earlier studies, collections were continued for several years (up to 5) and using hand netting methods—which is often associated with a higher rate of species detection than passive sampling methods [29], though rates of species detection in netting-based collections are presumably impacted by observer bias and skill [55]. However, bee abundances in the present study were similar to those found in both earlier works. The largest effects on bee diversity and FDis occurred early in the growing season (Fig 2), potentially indicating that species active primarily in spring have behavioral or life history traits that predispose them to site disturbance by livestock.

Collectively, our results have several implications for managers concerned with maintaining site occupancy by wild bee assemblages in rangelands where livestock production is a common land use. First, our results do not suggest that floral resources are enhanced in sites managed for cattle grazing as some earlier studies do. Neither did we find any evidence that grazed sites exhibited any reduction in floral resources, likely indicating that grazing practices (at the stocking levels controlled for here) in the region do not strongly impact bee

foraging habitats. Other recent studies indicate that increasing grazing intensities or higher stocking rates are generally associated with a reduction in available floral resources [56]. Here, floral resource availability was an important predictor of bee abundances. Second, bee assemblage composition did vary between grazed and non-grazed sites, and this was reflected by shifts in the ratios of *Bombus* spp: *Melissodes* spp. Further experimental work could help to elucidate whether this turnover in bee taxa is associated with variation in pollination services. Thirdly, both cattle grazing and high cheatgrass cover were associated with reduced site occupancy by above-ground nesting bees but increased site occupancy by below-ground nesting bees. Fourth, cattle grazing was associated with reduced FDis in early-season bee assemblages, and these effects may be mediated by cattle-driven impacts on nesting habitats rather than floral cover. Lastly, our study does not indicate that high cheatgrass cover is likely to negatively impact bee abundance or diversity, and may provide good nesting habitat. The mechanisms underlying this relationship are beyond the scope of the current study, but could have consequences for bee conservation, especially under widespread policies aimed at restoring cheatgrass-invaded habitats. For example, cheatgrass-dominated rangeland and forest sites are often treated with chemical [57], cultural [58], and physical [59] control methods with the general objective of reducing cheatgrass cover. Given that our study found an increased abundance of wild bees in cheatgrass sites, it will be important to determine whether cheatgrass control methods have deleterious, beneficial, or null impacts on bee assemblages to make appropriate management decisions about whether management of invasive grasses is likely to impact native bee conservation.

## Supporting information

**S1 Table. Bee functional traits considered in this study and their descriptions.** Functional trait values were determined from data or information found in Michener [33, 35, 37–40]. (DOCX)

**S2 Table. Functional trait values used to inform computation of bee assemblage functional dispersion.** For continuous variables (ITD), values are the mean from n = 10 specimens. (DOCX)

**S3 Table. Summary of bee trap captures by month of collection.** Months in which abundances were maximized were used to assign estimates of bee phenology for functional trait analysis. 'Early-season' = May or June; 'Mid-season' = July, and 'Late-season' = August. Cells are highlighted in gray to denote peak month of capture for each species. (DOCX)

**S4 Table. Summary of generalized linear model results to predict the effects of cover composition and floral richness on bee community assemblages.** Significant and marginally significant parameters are highlighted in bold text. (DOCX)

**S1 Fig. Species detection curves for wild bees in shortgrass prairie in three habitat types.** (DOCX)

## Acknowledgments

Samuel Murray, Fiona Horne and Haley Obermueller provided field and lab assistance. The staff of the C.P. Gillette Museum (Colorado State University) provided photography and storage/working space. We are also grateful to Virginia Scott (University of Colorado) and Dr.

Adrian Carper (University of Colorado), as well as Dr. Paul Rhoades (Idaho Department of Agriculture) for assistance with bee identification.

## Author Contributions

**Conceptualization:** Khum Bahadur Thapa-Magar, Thomas Seth Davis.

**Data curation:** Khum Bahadur Thapa-Magar, Boris Kondratieff.

**Formal analysis:** Khum Bahadur Thapa-Magar.

**Funding acquisition:** Thomas Seth Davis.

**Investigation:** Khum Bahadur Thapa-Magar, Boris Kondratieff.

**Project administration:** Thomas Seth Davis.

**Supervision:** Thomas Seth Davis.

**Validation:** Boris Kondratieff.

**Visualization:** Khum Bahadur Thapa-Magar.

**Writing – original draft:** Khum Bahadur Thapa-Magar.

**Writing – review & editing:** Thomas Seth Davis, Boris Kondratieff.

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
