## [Decision Letter · Decision Letter 0]

10 Sep 2020

PONE-D-20-23236

Livestock grazing is associated with seasonal reduction in pollinator biodiversity and functional dispersion but cheatgrass invasion is not: variation in bee assemblages in a multi-use shortgrass prairie

PLOS ONE

Dear Dr. Thomas Seth Davis

Thank you for submitting your manuscript to PLOS ONE. After careful consideration, we feel that it has merit but does not fully meet PLOS ONE’s publication criteria as it currently stands. Therefore, we invite you to submit a revised version of the manuscript that addresses the points raised during the review process.

We look forward to receiving your revised manuscript.

Kind regards,

Kleber Del-Claro, PhD

Academic Editor

PLOS ONE

Additional Editor Comments:

I strongly ask you to consider and answer the reviewers criticism, mainly : "The results are interesting, although it seems that the authors have not fully addressed their main finding, which seems to indicate that cheatgrass invaded sites support the highest bee numbers. It may also be that the sample size is too small to draw final conclusions about the effect of the habitat types studied. Somewhat problematic is the use of two habitat types that are not completely separated: Both non-grazed sites and sites invaded by cheatgrass are not grazed. The cheatgrass category is simply a subcategory of non-grazed sites, which is also indicated by NMDS, which shows the highest similarity between cheatgrass assemblages. " I suggest you to answer it clearly.

2. In your Methods section, please provide additional location information of the study sites, including geographic coordinates for the data set if available.

4. Please ensure that you refer to Figure 1 in your text as, if accepted, production will need this reference to link the reader to the figure.

5. Please include captions for your Supporting Information files at the end of your manuscript, and update any in-text citations to match accordingly. Please see our Supporting Information guidelines for more information: http://journals.plos.org/plosone/s/supporting-information

Reviewers' comments:

Reviewer's Responses to Questions

**Comments to the Author**

1. Is the manuscript technically sound, and do the data support the conclusions?

Reviewer #1: Yes

Reviewer #2: Yes

2. Has the statistical analysis been performed appropriately and rigorously? 

Reviewer #1: Yes

Reviewer #2: Yes

3. Have the authors made all data underlying the findings in their manuscript fully available?

Reviewer #1: Yes

Reviewer #2: Yes

4. Is the manuscript presented in an intelligible fashion and written in standard English?

Reviewer #1: Yes

Reviewer #2: Yes

5. Review Comments to the Author

Reviewer #1: The manuscript “Livestock grazing is associated with seasonal reduction in pollinator biodiversity and functional dispersion but cheatgrass invasion is not: variation in bee assemblages in a multi-use shortgrass prairie” by Khum Bahadur Thapa-Magar and colleagues deals with an important and current topic, the role of grazing and the presence of invasive grass (cheatgrass - Bromus tectorum and B. japonicum) on bee assemblages.

The manuscript is well written and the data are analyzed in a meaningful way. The study indicates that cattle grazing has a more negative impact on bee abundance and diversity than the increased occurrence of invasive cheatgrass. On the contrary, the sites dominated by cheatgrass even seemed to promote the highest number of bee individuals and species. In addition to the effects of the habitat categories (1) cattle pasture, (2) non-grazed sites dominated by cheatgrass, and (3) non-grazed sites without cheatgrass, other environmental variables related to foraging (e.g. flower cover) and nesting (e.g. bare ground cover and wood/litter material) were analyzed and were related to bee diversity and assemblage characteristics.

The following points can be considered to further improve the manuscript:

General comments:

1. since it is of central importance for the entire study, the definition and naming of the three habitat (site) categories must be clarified. As I understood, the two categories "cheatgrass-colonized" and "non-grazed" are not grazed by cattle. However, the naming and consideration in many parts of the manuscript leave the impression that "non-grazed" is a unique feature. It rather seems that "cheatgrass-colonized" is only a subcategory of "non-grazed". This has to be addressed more clearly throughout the manuscript (also in the presentation and interpretation of the results). For example, it is also indicated in the results, where NMDS indicates that "cheatgrass assemblages" are most similar to each other...

2. it seems that 'cheatgrass sites' support the highest number of bee individuals and species (also compared to non-grazed sites without cheatgrass). This finding would be interesting, but it is not clear (a) whether there is a statistically significant difference, as paired testing between habitats in terms of numbers and species richness seems to be missing and (b) how this difference could be explained. Here it might be helpful to consider the role of the different habitats either for foraging or for nesting.

3. if no differences in bee numbers between habitat types can be demonstrated, the authors should indicate whether the sample size (n=10) might be too small for final conclusions.

Specific comments:

l. 21: nesting habitats: what about bare soil as nesting habitat for below-ground nesting bees?

l. 25: how was bee diversity in the late season compared to early/peak season? Is a reduced difference explained by reduction of bee species richness? Are only some very generalistic bees left at the end of the season?

l. 33:…and our STUDY suggests… (add “study”)

l. 74: which “mechanisms” seem probable? Could be outlined at least a little; e.g. replacement of forbs that are used as foraging or nesting resource by grasses that may not provide food or useful nest-site structures…

l. 82: functional traits not explicitly introduced: it should be mentioned before why foraging and nesting resources may be linked to certain functional traits

l. 98: are grasses in natural (conservation) areas grazed by any mammalian herbivore? Larger mammalian herbivores? Otherwise ‘non-grazed’ may represent a more “artificial, manmade” situation than ‘grazed’ by suitable cattle densities, as in a “natural” situation grazing by bison, elk, pronghorn etc. would at least temporarily occur. This aspect might be considered also for the interpretation of the data.

l. 103 or elsewhere: the size or size range of patches of the different habitat types need to be mentioned. Are patches so small that traps might be attracting bees from different habitat types, or are all bees in a trap based in the specific habitat? Is it possible to state whether bees collected in a specific habitat used the habitat rather for foraging or for nesting?

l. 113: method to quantify floral cover might be explained in some more detail. What does it mean ‘proportion of forb contacts…’?

l. 142-144: body size traits of 10 specimens of 49 species: is it correct that no rare species with less than 10 individuals (5f, 5m) were found? This seems to be remarkable and might be addressed in the discussion.

l. 188 and other places: what are bee community assemblages? Restrict to “bee assemblage” or “bee community”. I would suggest ‘assemblage’, as collected bees are probably no community in the strict sense (check definition of assemblage vs. community in the context of groups of interacting organisms…).

l. 210: it should be explained why it is meaningful to analyse the question whether non-grazed sites have lower cheatgrass cover than cheat-grass sites. Is this a result or rather should be shown in materials/methods?

l. 225-232: it should be shown more explicitly which habitat categories are different. Based on Figure 2, ‘cheatgrass’ seems to support the highest number of bee individuals and bee species. Is this difference significant and if yes, to different to which category?

l. 258-280: this section provides many results without clear coverage by statistical tests. It needs to be explained whether and how all these results are derived from Figure 5, or further tests should be provided.

l. 303: is it high floral cover or type/species of flowering plant? Preponderance of bumble bees in sites characterised by high floral cover indicates/suggests that not only floral availability but also specific characteristics of flowers might be different between these sites. Can information on plant taxa be provided and used for discussion whether plant identity might have an explanatory potential?

l. 348: the current study has not (systematically) compared different grazing intensities of cattle (as suggested by the reference to intermediate grazing as the best prerequisite for high availability of floral resources); it cannot therefore make any statements about the role of grazing intensity, which is likely to have a strong impact…

l. 360: as cattle density / grazing intensity is highly important for effects of grazing on flower availability, this always needs to be considered. Can comparison/discussion for different cattle densities be provided?

Reviewer #2: For this work it is important to use a passive method, which equalize the effort. So, I understand that the use of a trap is essencial. But, since this blue vane trap is not widespread among the bee biologists, please provide some comments about the equal efficient of this trap and if there is a bias to any bee group (as occurs in pantraps).

For Table 2 I suggest to include the 3 habitat types and then provide the abundance of the species for each.

Please, revise the traits or the interpretation assumed to the cleptoparasite bees since they can attack different bee species (solitary (most probably) and social), they eat the avaliable pollen they found in the brood cell (being oligoletic or polyletic), visit different plants for nectar and they do not collect pollen (for sure no scopa).

6. PLOS authors have the option to publish the peer review history of their article (what does this mean?). If published, this will include your full peer review and any attached files.

Reviewer #1: No

Reviewer #2: No

---

## [Author Response · Author response to Decision Letter 0]

12 Oct 2020

Editor Comments:

I strongly ask you to consider and answer the reviewers criticism, mainly : "The results are interesting, although it seems that the authors have not fully addressed their main finding, which seems to indicate that cheatgrass invaded sites support the highest bee numbers. It may also be that the sample size is too small to draw final conclusions about the effect of the habitat types studied. Somewhat problematic is the use of two habitat types that are not completely separated: Both non-grazed sites and sites invaded by cheatgrass are not grazed. The cheatgrass category is simply a subcategory of non-grazed sites, which is also indicated by NMDS, which shows the highest similarity between cheatgrass assemblages. " I suggest you to answer it clearly.

>We have carefully considered these comments and provide a detailed response to Reviewer 1 below, in addition to presenting new material to address the comments. 

>Done.

2. In your Methods section, please provide additional location information of the study sites, including geographic coordinates for the data set if available.

>Approximate coordinates are shown in Figure 1. Since some of our sites are private lands, we did not feel it appropriate to share coordinates publicly. 

>Done. 

4. Please ensure that you refer to Figure 1 in your text as, if accepted, production will need this reference to link the reader to the figure.

>Done.

 5. Please include captions for your Supporting Information files at the end of your manuscript, and update any in-text citations to match accordingly. Please see our Supporting Information guidelines for more information: http://journals.plos.org/plosone/s/supporting-information

>Done.

Reviewer #1

1. since it is of central importance for the entire study, the definition and naming of the three habitat (site) categories must be clarified. As I understood, the two categories "cheatgrass-colonized" and "non-grazed" are not grazed by cattle. However, the naming and consideration in many parts of the manuscript leave the impression that "non-grazed" is a unique feature. It rather seems that "cheatgrass-colonized" is only a subcategory of "non-grazed". This has to be addressed more clearly throughout the manuscript (also in the presentation and interpretation of the results). For example, it is also indicated in the results, where NMDS indicates that "cheatgrass assemblages" are most similar to each other...

>We agree with the reviewers comment and have made it clearer in the Methods section and Discussion section that we considered 20 non-grazed sites, which were selected on the basis of exhibiting either (1) low cheatgrass cover or (2) high cheatgrass cover. In figures and text we now refer to our categories as ‘cattle grazed’ sites, and non-grazed sites with ‘low cheatgrass cover’ and ‘high cheatgrass cover’—we have tried to be consistent with this terminology throughout the manuscript. Also, we now move presentation of Table 1 to reflect a methodological confirmation/validation that cheatgrass cover and native grass cover differed between habitat types, rather than treating this as a stand-alone objective and result.

Regarding interpretation of the NMDS: while considering the reviewer comment it was realized that a mistake was made in the labelling of ellipses—we have switched the labels for sites with low cheatgrass cover and high cheatgrass cover, which changes the interpretation of the result (ie, bee assemblages in sites with low cheatgrass cover are the most diverse). We have altered the text to reflect this difference. 

2. it seems that 'cheatgrass sites' support the highest number of bee individuals and species (also compared to non-grazed sites without cheatgrass). This finding would be interesting, but it is not clear (a) whether there is a statistically significant difference, as paired testing between habitats in terms of numbers and species richness seems to be missing and (b) how this difference could be explained. Here it might be helpful to consider the role of the different habitats either for foraging or for nesting.

>(a) We discuss the statistical significance of a two-way ANOVA model examining main effects of habitat type (cattle-grazed sites and non-grazed sites with low or high cheatgrass cover), collection month, and the habit x collection month interaction on bee abundance, richness, diversity, and functional dispersion between L220-250. We also now more explicitly discuss the post-hoc tests in the Results section and the % differences between least-squared means, and have compiled a new table (Table 3) to show sample means, errors, and pairwise comparisons of means groupings (Tukey’s HSD test) for clarity.

(b) We attempt to provide some explanation/hypotheses about why this pattern (higher early-season bee abundance and diversity in sites with high cheatgrass cover) may have occurred at L342-361. However we are concerned of being overly-speculative about this result and chose to cautiously interpret our data to indicate that ‘cheatgrass does not have negative effects on bee assemblages’ rather than concluding that ‘cheatgrass positively impacts bee assemblages’, and we make a call for further research into potential mechanisms. 

3. if no differences in bee numbers between habitat types can be demonstrated, the authors should indicate whether the sample size (n=10) might be too small for final conclusions.

>Please see response to section (a) above. It seems that Figure 2a, Table 3, and the results described at L220-229 essentially address this comment. 

Specific comments:

l. 21: nesting habitats: what about bare soil as nesting habitat for below-ground nesting bees?

>’bare soil’ has been added here. 

l. 25: how was bee diversity in the late season compared to early/peak season? Is a reduced difference explained by reduction of bee species richness? Are only some very generalistic bees left at the end of the season?

>We now mention that Melissodes spp .and Bombus spp. appear to become much more prominent as the season progresses, which likely drives these patterns. 

l. 33:…and our STUDY suggests… (add “study”)

>Done. 

l. 74: which “mechanisms” seem probable? Could be outlined at least a little; e.g. replacement of forbs that are used as foraging or nesting resource by grasses that may not provide food or useful nest-site structures…

>Additional comments suggested by the reviewer have been added here. 

l. 82: functional traits not explicitly introduced: it should be mentioned before why foraging and nesting resources may be linked to certain functional traits

>A brief introduction on the potential role of functional variation is now given at L66-68.

l. 98: are grasses in natural (conservation) areas grazed by any mammalian herbivore? Larger mammalian herbivores? Otherwise ‘non-grazed’ may represent a more “artificial, manmade” situation than ‘grazed’ by suitable cattle densities, as in a “natural” situation grazing by bison, elk, pronghorn etc. would at least temporarily occur. This aspect might be considered also for the interpretation of the data.

>To our knowledge, ungulates such as elk and pronghorn are not excluded from sites designated as ‘non-grazed’. We now specify at L107-108 that ‘non-grazed’ refers to cattle grazing and that wild ungulates are not excluded from these areas. 

l. 103 or elsewhere: the size or size range of patches of the different habitat types need to be mentioned. Are patches so small that traps might be attracting bees from different habitat types, or are all bees in a trap based in the specific habitat? Is it possible to state whether bees collected in a specific habitat used the habitat rather for foraging or for nesting?

> Although we agree that this is an interesting point, it was somewhat beyond the scope of our study to determine whether collected bees were responding to patch size or landscape factors. So, although we realize this is an important aspect of bee community ecology, we feel it would be beyond the limits of our data to speculate on this point given our study design. However, we have added a brief mention of this point as a potential explanation for why higher bee abundances were observed in sites with high cheatgrass cover at L347-350.

l. 113: method to quantify floral cover might be explained in some more detail. What does it mean ‘proportion of forb contacts…’?

>We have rephrased to use the term ‘intercepts’ and to more clearly explain that transects were point-intercept transects. 

l. 142-144: body size traits of 10 specimens of 49 species: is it correct that no rare species with less than 10 individuals (5f, 5m) were found? This seems to be remarkable and might be addressed in the discussion.

>Museum specimens were used to derive trait values for rare species. This is now mentioned in the methodology. 

l. 188 and other places: what are bee community assemblages? Restrict to “bee assemblage” or “bee community”. I would suggest ‘assemblage’, as collected bees are probably no community in the strict sense (check definition of assemblage vs. community in the context of groups of interacting organisms…).

> We have corrected to ‘bee assemblage’ or ‘bee community’ throughout as appropriate. 

l. 210: it should be explained why it is meaningful to analyse the question whether non-grazed sites have lower cheatgrass cover than cheat-grass sites. Is this a result or rather should be shown in materials/methods?

> We have moved this result and Table 1 as a validation of methodology instead of treating this as a stand-alone objective and result. Accordingly, the original Objective 1 has been deleted and merged into the Methods section. 

l. 225-232: it should be shown more explicitly which habitat categories are different. Based on Figure 2, ‘cheatgrass’ seems to support the highest number of bee individuals and bee species. Is this difference significant and if yes, to different to which category?

>This should now be addressed by the addition of Table 3. 

l. 258-280: this section provides many results without clear coverage by statistical tests. It needs to be explained whether and how all these results are derived from Figure 5, or further tests should be provided.

> All shaded cells shown in Figure 5 are ‘significant’ correlations at the P<0.10 level; therefore most articles using this method do not exhaustively provide all p values and correlation metrics—instead, the typical approach is to provide a model deviance and overall model significance as we have done at L264, and to show the patterns of significance graphically as we do in Figure 5. 

l. 303: is it high floral cover or type/species of flowering plant? Preponderance of bumble bees in sites characterised by high floral cover indicates/suggests that not only floral availability but also specific characteristics of flowers might be different between these sites. Can information on plant taxa be provided and used for discussion whether plant identity might have an explanatory potential?

>We are using floral cover here, rather than floral community composition. 

l. 348: the current study has not (systematically) compared different grazing intensities of cattle (as suggested by the reference to intermediate grazing as the best prerequisite for high availability of floral resources); it cannot therefore make any statements about the role of grazing intensity, which is likely to have a strong impact…

> We agree with the reviewers’ point and have deleted the reference to our study in this context. 

l. 360: as cattle density / grazing intensity is highly important for effects of grazing on flower availability, this always needs to be considered. Can comparison/discussion for different cattle densities be provided?

> We agree with the reviewer and have provided a caveat and additional reference to this point here. 

Reviewer #2: 

For this work it is important to use a passive method, which equalize the effort. So, I understand that the use of a trap is essencial. But, since this blue vane trap is not widespread among the bee biologists, please provide some comments about the equal efficient of this trap and if there is a bias to any bee group (as occurs in pantraps).

>A few comments are now provided to this effect at L372-376

For Table 2 I suggest to include the 3 habitat types and then provide the abundance of the species for each.

>Done. Also, this edit makes the original Table S4 (summary of bee genera by habitat classification) redundant—accordingly, the original Table S4 has been removed from the ms and is replaced with the previous Table S5 (glm summary). 

Please, revise the traits or the interpretation assumed to the cleptoparasite bees since they can attack different bee species (solitary (most probably) and social), they eat the avaliable pollen they found in the brood cell (being oligoletic or polyletic), visit different plants for nectar and they do not collect pollen (for sure no scopa).

> Entries for kleptoparasotic species have been updated in Table S1 and S2 as per the reviewer’s request.

---

## [Decision Letter · Decision Letter 1]

19 Nov 2020

Livestock grazing is associated with seasonal reduction in pollinator biodiversity and functional dispersion but cheatgrass invasion is not: variation in bee assemblages in a multi-use shortgrass prairie

PONE-D-20-23236R1

Dear Dr. Thomas Seth Davis,

We’re pleased to inform you that your manuscript has been judged scientifically suitable for publication and will be formally accepted for publication once it meets all outstanding technical requirements.

Kind regards,

Kleber Del-Claro, PhD

Academic Editor

PLOS ONE

Additional Editor Comments (optional):

Reviewers' comments:

Reviewer's Responses to Questions

**Comments to the Author**

1. If the authors have adequately addressed your comments raised in a previous round of review and you feel that this manuscript is now acceptable for publication, you may indicate that here to bypass the “Comments to the Author” section, enter your conflict of interest statement in the “Confidential to Editor” section, and submit your "Accept" recommendation.

Reviewer #1: All comments have been addressed

Reviewer #2: All comments have been addressed

2. Is the manuscript technically sound, and do the data support the conclusions?

Reviewer #1: (No Response)

Reviewer #2: Yes

3. Has the statistical analysis been performed appropriately and rigorously? 

Reviewer #1: (No Response)

Reviewer #2: Yes

4. Have the authors made all data underlying the findings in their manuscript fully available?

Reviewer #1: (No Response)

Reviewer #2: Yes

5. Is the manuscript presented in an intelligible fashion and written in standard English?

Reviewer #1: (No Response)

Reviewer #2: Yes

6. Review Comments to the Author

Reviewer #1: (No Response)

Reviewer #2: (No Response)

7. PLOS authors have the option to publish the peer review history of their article (what does this mean?). If published, this will include your full peer review and any attached files.

Reviewer #1: No

Reviewer #2: No

---

## [Editor Report · Acceptance letter]

1 Dec 2020

PONE-D-20-23236R1 

Livestock grazing is associated with seasonal reduction in pollinator biodiversity and functional dispersion but cheatgrass invasion is not: variation in bee assemblages in a multi-use shortgrass prairie 

Dear Dr. Davis:

I'm pleased to inform you that your manuscript has been deemed suitable for publication in PLOS ONE. Congratulations! Your manuscript is now with our production department. 

Kind regards, 

on behalf of

Dr. Kleber Del-Claro 

Academic Editor

PLOS ONE